# Bridging Imagination and Reality for Model-Based Deep Reinforcement Learning

**Guangxiang Zhu**[*]
IIIS
Tsinghua University
guangxiangzhu@outlook.com

**Minghao Zhang**[*]
School of Software
Tsinghua University
mehoozhang@gmail.com

**Honglak Lee**
EECS
University of Michigan
honglak@eecs.umich.edu

**Chongjie Zhang**
IIIS
Tsinghua University
chongjie@tsinghua.edu.cn

## Abstract

Sample efficiency has been one of the major challenges for deep reinforcement learning. Recently, model-based reinforcement learning has been proposed to address this challenge by performing planning on imaginary trajectories with a learned world model. However, world model learning may suffer from overfitting to training trajectories, and thus model-based value estimation and policy search will be prone to be sucked in an inferior local policy. In this paper, we propose a novel model-based reinforcement learning algorithm, called **BrI**dging **R**eality and **D**ream (**BIRD**). It maximizes the mutual information between imaginary and real trajectories so that the policy improvement learned from imaginary trajectories can be easily generalized to real trajectories. We demonstrate that our approach improves sample efficiency of model-based planning, and achieves state-of-the-art performance on challenging visual control benchmarks.

## 1 Introduction

Reinforcement learning (RL) is proposed as a general-purpose learning framework for artificial intelligence problems, and has led to tremendous progress in a variety of domains [1, 2, 3, 4]. Model-free RL adopts a trail-and-error paradigm, which directly learns a mapping function from observations to values or actions through interactions with environments. It has achieved remarkable performance in certain video games and continuous control tasks because of its simplicity and minimal assumptions about environments. However, model-free approaches are not yet sample efficient and require several orders of magnitude more training samples than human learning, which limits its applications on real-world tasks [5].

A promising direction for improving sample efficiency is to explore model-based RL, which first builds an action-conditioned world model and then performs planning or policy search based on the learned model. The world model needs to encode the representations and dynamics of an environment is then used as a "dreamer" to do multi-step lookaheads for planning or policy search. Recently, world models based on deep neural networks were developed to handle dynamics in complex high-dimensional environments, which offers opportunities for learning model-based polices with visual observations [6, 7, 8, 9, 10, 11, 12, 13].

---

[*]Equal Contribution

Model-based frameworks can be roughly grouped into four categories. First, Dyna-style algorithms alternate between building the world model from interactions with environments and performing policy optimization on simulated data generated by the learned model [14, 15, 16, 17, 11]. Second, model predictive control (MPC) and shooting algorithms alternate model learning, planning and action execution [18, 19, 20]. Third, model-augmented value expansion algorithms use model-based rollouts to improve targets for model-free temporal difference (TD) updates or policy gradients [21, 9, 6, 10]. Fourth, analytic-gradient algorithms leverage the gradients of the model-based imaginary returns with respect to the policy and directly propagate such gradients through a differentiable world model to the policy network [22, 23, 24, 25, 26, 27, 13]. Compared to conventional planning algorithms that generate numerous rollouts to select the highest performing action sequence, analytic-gradient algorithm is more computationally efficient, especially in complex domains with deep neural networks. Dreamer [13] as a landmark of analytic-gradient model-based RL, achieves state-of-the-art performance on visual control tasks.

However, most existing breakthroughs on analytic gradients focus on optimizing the policy on imaginary trajectories and leave the discrepancy between imagination and reality largely unstudied, which often bottlenecks their performance on real trajectories. In practice, a learning-based world model is not perfect, especially in complex environments. Unrolling with an imperfect model for multiple steps generates a large accumulative error, leaving a gap between the generated trajectories and reality. If we directly optimize policy based on the analytic gradients through the imaginary trajectories, the policy will tend to deviate from reality and get sucked in an inferior local solution.

Evidence from humans' cognition and learning in the physical world suggests that humans naturally have the capacity of self-reflection and introspection. In everyday life, we track and review our past thoughts and imaginations, introspect to further understand our internal states and interactions with the external world, and change our values and behavior patterns accordingly [28, 29]. Inspired by this insight, our basic idea is to leverage information from real trajectories to endow policy improvement on imaginations with awareness of discrepancy between imagination and reality. We propose a novel reality-aware model-based framework, called **BrI**dging **R**eality and **D**ream (**BIRD**), which performs differentiable planning on imaginary trajectories, as well as enables adaptive generalization to reality for learned policy by optimizing mutual information between imaginary and real trajectories. Our model-based policy optimization framework naturally unifies confidence-aware analytic gradients, entropy regularization maximization, and model learning. We conduct experiments on challenging visual control benchmarks (DeepMind Control Suite with image inputs [30]) and the results demonstrate that BIRD achieves state-of-the-art performance in terms of sample efficiency. Our ablation study further verifies the superiority of BIRD benefits from mutual information maximization rather than from the increase of policy entropy.

## 2 Related Work

**Model-Based Reinforcement Learning**     Model-based RL exhibits high sample efficiency and has been widely used in several real-world control tasks, such as robotics [31, 32, 7]. Dyna-style algorithms [14, 15, 16, 17, 11] optimize policies with samples generated from a learned world model. Model predictive control (MPC) and shooting methods [18, 19, 20] leverage planning to select actions, but suffer from expensive computation. In model-augmented value expansion approaches, MVE [21], VPN [6] and STEVE [9] use model-based rollouts to improve targets for model-free TD updates. MuZero [10] further incorporates Monte-Carlo tree search (MCTS) and achieves remarkable performance on Atari and board games. To manage visual control tasks, VisualMPC [33] introduces a visual prediction model to keep track of entities through occlusion by temporal skip connections. PlaNet [12] improves the model learning by combining deterministic and stochastic latent dynamics models. [34] presents a summary of model-based approaches and benchmarks popular algorithms for comparisons and extensions.

**Analytic Value Gradients**     If a differentiable world model is available, analytic value gradients are proposed to directly update the policy by gradients that flow through the world model. PILCO [24] and iLQR [25] compute an analytic gradient by assuming Gaussian processes and linear functions for the dynamics model, respectively. Guided policy search (GPS) [26, 35, 36, 37, 38] uses deep neural networks to distill behaviors from the iLQR controller. Value Gradients (VG) [22] and Stochastic Value Gradients (SVG) [23] provide a new direction to calculate analytic value gradients through a generic differentiable world model. Dreamer [13] and IVG [27] further extend SVG by

generating imaginary rollouts in the latent space. However, these works focus on improving the policy in imaginations, leaving the discrepancy between imagination and reality largely unstudied. Our approach enables policy generalization to real-world interactions by maximizing mutual information between imagination and real trajectories, while optimizing the policy on imaginary trajectories. In addition, alternative end-to-end planning methods [39, 40] leverage analytic gradients, but they either focus on online planning in simple tasks [39] or require goal images and distance metrics for the reward function [40].

**Information-Based Optimization**    In addition to maximizing the expected return objective, a reliable RL agent may exhibit more characteristics, like meaningful representations, strong generalization, and efficient exploration. Deep information-based methods [41, 42, 43, 44] recently show progress towards this direction. [45, 46, 47] are proposed to learn more efficient representations. Maximum entropy RL maximizes the entropy regularized return to obtain a robust policy [48, 49] and [50, 51] further connect policy optimization under such regularization with value based RL. [52] learns a goal-conditioned policy with information bottleneck to identify decision states. IDS [53] estimates the information gain for a sampling-based exploration strategy. These algorithms mainly focus on facilitating policy learning in the model-free setting, while BIRD aims at bridging imagination and reality by mutual information maximization in the context of model-based RL.

## 3    Preliminaries

### 3.1    Reinforcement Learning

A reinforcement learning agent aims at learning a policy to maximize the cumulative rewards by exploring in a Markov Decision Processes (MDP) [54]. Normally, we use denote time step as $t$ and introduce state $s_t \in \mathcal{S}$, action $a_t \in \mathcal{A}$, reward function $r(s_t, a_t)$, a policy $\pi_\theta(s)$, and a transition probability $p(s_{t+1}|s_t, a_t)$ to characterize the process of interacting with the environment. The goal of the agent is to find a policy parameter $\theta$ that maximizes the long-horizon summed rewards represented by a value function $v_\phi(s_t) \doteq \mathbb{E}\left(\sum_{i=t}^{t+H} \gamma^{i-t} r_i\right)$ parameterized with $\phi$. In model-based RL, the agent builds a world model $p_\psi$ parameterized by $\psi$ for environmental dynamics $p$ and reward function $r$, and then performs planning or policy search based on this model.

### 3.2    World Model

Considering that several complex tasks (e.g., visual control tasks [30]) are partially observable Markov decision process (POMDP), this paper adopts a similar world model with PlaNet [12] and Dreamer [13], which learns latent states from the history of visual observations and models the latent dynamics by LSTM-like recurrent networks. Specifically, the world model consists of the following modules:

$$
\begin{aligned}
\text{Representation model}: \quad & s_t \sim p_\psi(s_t|s_{t-1}, a_{t-1}, o_t) \\
\text{Transition model}: \quad & s_t \sim p_\psi(s_t|s_{t-1}, a_{t-1}) \\
\text{Observation model}: \quad & o_t \sim p_\psi(o_t|s_t) \\
\text{Reward model}: \quad & r_t \sim p_\psi(r_t|s_t).
\end{aligned}
\tag{1}
$$

The representation model encodes the image input into a compact latent space and the long-horizon dynamics on latent states are captured by a latent transition model. We use RSSM [12] as our transition model, which combines deterministic and stochastic transition model in order to learn dynamics more accurately and efficiently. For each latent state on the predicted trajectories, observation model learns to reconstruct its visual observations, and the reward model predicts the immediate reward. The entire world model $\mathcal{J}_\psi^{\text{Model}}$ is optimized by a VAE-like objective [55]:

$$
\begin{aligned}
\mathcal{J}_\psi^{\text{Model}}(\tau^{\text{img}}, \tau^{\text{real}}) = \sum_{(a_{t-1}, o_t, r_t) \sim \tau^{\text{real}}} & \Big[ \ln(p_\psi(o_t|s_t)) + \ln(p_\psi(r_t|s_t)) \\
& - \beta D_{\text{KL}}(p_\psi(s_t|s_{t-1}, a_{t-1}, o_t)||p_\psi(s_t|s_{t-1}, a_{t-1})) \Big].
\end{aligned}
\tag{2}
$$

### 3.3 Stochastic Value Gradients

Given a differentiable world model, stochastic value gradients (SVG) [22, 23] can be applied to directly compute policy gradient on the whole imaginary trajectory, which is a recursive composition of policy, transition, reward, and value function. According to the stochastic Bellman Equation, we have:

$$v(s) = \mathbb{E}_{\rho(\eta)} \left( r(s, \pi_\theta(s, \eta)) + \gamma \mathbb{E}_{\rho(\xi)} \left( v(p(s, \pi_\theta(s, \eta), \xi)) \right) \right), \tag{3}$$

where $\eta \sim \rho(\eta)$ and $\xi \sim \rho(\xi)$ are noises from a fixed noise distribution for re-parameterization. So the gradients through trajectories can be iteratively computed as:

$$\frac{\partial v}{\partial s} = \mathbb{E}_{\rho(\eta)} \left( \frac{\partial \mathrm{r}}{\partial s} + \frac{\partial r}{\partial a} \frac{\partial \pi}{\partial s} + \gamma \mathbb{E}_{\rho(\xi)} \left( \frac{\partial v}{\partial s'} \left( \frac{\partial p}{\partial s} + \frac{\partial p}{\partial a} \frac{\partial \pi}{\partial s} \right) \right) \right)$$

$$\frac{\partial v}{\partial \theta} = \mathbb{E}_{\rho(\eta)} \left( \frac{\partial r}{\partial a} \frac{\partial \pi}{\partial \theta} + \gamma \mathbb{E}_{\rho(\xi)} \left( \frac{\partial v}{\partial s'} \frac{\partial p}{\partial a} \frac{\partial \pi}{\partial \theta} + \frac{\partial v}{\partial \theta} \right) \right), \tag{4}$$

where $s'$ denotes the next state given by the transition function. Intuitively, policy can be improved by propagating analytic gradients with respect to the policy network through the imaginary trajectories.

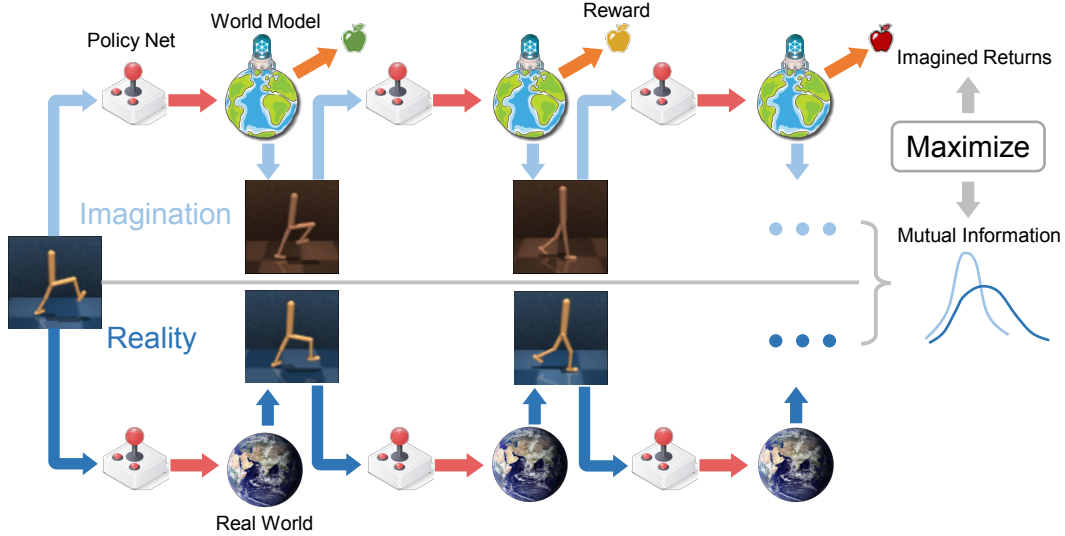

Figure 1: Overall framework of BIRD.

## 4 Reality-Aware Model-Based Policy Improvement

In this section, we present a novel model-based RL framework, called **BrIdging Reality and Dream** (**BIRD**), as shown in Figure 1. The agent represents its policy function with a policy network ( ). To estimate the future effects of its policy and enable potential policy improvement, it unrolls trajectories based on its world model ( ) using the current policy and optimizes the accumulative rewards on the imaginary trajectories. The policy network and differentiable world model connect to one another forming a larger trainable network, which supports differentiable planning and allows the analytic gradients of accumulative rewards with respect to the policy flow through the world model. In the meantime, the agent also interacts with the real world ( ) and generates real trajectories. BIRD maximizes the mutual information between real and imaginary trajectories to endow both the policy network and the world model with adaptive generalization to real-world interactions. In summary, BIRD maximizes the total objective function:

$$\mathcal{J}_{\mathrm{BIRD}} = \mathcal{J}_\theta^{\mathrm{SVG}}(\tau^{\mathrm{img\_roll}}) - \mathcal{L}_\phi^{\mathrm{TD}}(\tau^{\mathrm{img\_roll}}) + w I_{\theta, \psi}(\tau^{\mathrm{img}}, \tau^{\mathrm{real}}), \tag{5}$$

where $\tau^{\mathrm{real}}$ and $\tau^{\mathrm{img}}$ indicate the real trajectories and the corresponding imaginary trajectories under the same policy, and $\tau^{\mathrm{img\_roll}}$ indicate the rolled out imaginary trajectories during the optimization

of policy improvement. $\theta, \phi, \psi$, are parameters of policy network $\pi_\theta$, value network $v_\phi$, and world model $p_\psi$, respectively. The first two terms $\mathcal{J}_\theta^{\text{SVG}}(\tau^{\text{img\_roll}}) - \mathcal{J}_\phi^{\text{TD}}(\tau^{\text{img\_roll}})$ account for policy improvement on imaginations, the last term $I_{\theta,\psi}(\tau^{\text{img}}, \tau^{\text{real}})$ optimizes the mutual information, and $w$ is a weighting factor between them.

In conventional model-based RL approaches, real-world trajectories are normally used to optimize model prediction error, which is quite different from BIRD. In complex domains, optimizing model prediction error cannot guarantee a perfect predictive model. Unrolling with such an imperfect model for multiple steps will generate a large accumulative error, leaving a large gap between the generated trajectories and real ones. Thus, policy optimized by such a model may overfit undesirable imaginations and have a low generalization ability to reality, which is also shown in our experiments (Figure 3). This problem is further exacerbated in analytic-gradient RL that performs differentiable planning by gradient-based local search. This is because even a small gradient step along the imperfect model can easily reach a non-generalizable neighbourhood and lead to a direction of incorrect policy improvement. To address this problem, our method optimizes mutual information with respect to both the model and the policy, which makes policy improvement aware of the discrepancy between real and imaginary trajectories. Intuitively, BIRD optimizes the world model to be more real and reinforces the actions whose resulting imaginations not only have large accumulative rewards, but also resemble real trajectories. As a result, BIRD learns a policy from imaginations with easier generalization to the real-world environment.

## 4.1 Policy Improvement on Imaginations

As a model-based RL algorithm, BIRD improves the policy by maximizing the accumulative rewards of the imaginary trajectories unrolled by the world model. Conventional model-based approaches [18, 7, 11] perform policy improvement by selecting the optimal action sequence that maximizes the expected planning reward, that is $\max_{a_{t:t+H}} \mathbb{E}_{s_x \sim p_\psi} \sum_{x=t}^{t+H} r(s_x, a_x)$. If the world model is differentiable, we use stochastic value gradients (SVG) to directly leverage the gradients through the world model for policy improvement. Similar with Dreamer [13], our objective of maximizing the model-based value expansion within horizon $H$ is given by:

$$\mathcal{J}_\theta^{\text{SVG}}(\tau^{\text{img}}) = \max_\theta \sum_{x=t}^{t+H} \text{V}_\lambda(s_x),$$

$$\text{V}_\lambda(s_x) = \mathbb{E}_{a_i \sim \pi_\theta, s_i \sim p_\psi(s_i|s_{i-1},a_{i-1})} \sum_{k=1}^{H} \lambda_k \left[ \left( \sum_{i=t}^{h-1} \gamma^{i-t} r_i \right) + \gamma^{h-t} v_\phi(s_h) \right], \quad (6)$$

where $r_i$ represents the immediate reward at timestep $i$ predicted by the world model $\psi$. For each expand length $k$, we expand the expected value from current timestep $x$ to timestep $h - 1$ ($h = \min(x + k, t + H)$) and use learned value function $v_\phi(s_h)$ to estimate returns beyond $h$ steps, i.e., $v_\phi(s_h) = \mathbb{E}\left( \sum_{i=h}^{H} \gamma^{i-h} r_i \right)$. Here, we use the exponentially-weighted average of the estimates for different values of $k$ to balance bias and variance, and the exponential weighting factor is indicated by $\lambda_k$. As shown in the Equation 6, we alternate the policy network $\pi_\theta$ and the differentiable world model $p_\psi$, connect them to one another to form a large end-to-end trainable network, and then back-propagate the gradients of expected values with respect to policy parameters $\theta$ though this large network. Intuitively, a gradient step of the policy network encourages the world model to obtain a gradient step of new states, and in turn affect the future value. As a result, the states and policy will be optimized sequentially based on the feedback on future values. To optimize the value network, we use TD updates as actor-critic algorithms [54, 56, 21], instead of Monte Carlo estimation:

$$\mathcal{L}_\phi^{\text{TD}}(\tau^{\text{img}}) = \sum_{x=t}^{t+H} \|v_\phi(s_x) - \text{V}_\lambda(s_x)\|^2, \quad (7)$$

## 4.2 Bridge Imagination and Reality by Mutual Information Maximization

To ensure the policy improvement based on the learned world model is equally effective in the real world, we introduce an information-theoretic objective, that optimizes mutual information between

real and imaginary trajectories with respect to the policy network and the world model:

$$
\begin{aligned}
I_{\theta,\psi}(\tau^{\text{img}}, \tau^{\text{real}}) &= \mathcal{H}(\tau^{\text{real}}) - \mathcal{H}(\tau^{\text{real}}|\tau^{\text{img}}) \\
&= \mathcal{H}(\tau^{\text{real}}) - \sum_u P(u)\mathcal{H}(\tau^{\text{real}}|\tau^{\text{img}} = u) \\
&= \mathcal{H}(\tau^{\text{real}}) + \sum_u P(u) \sum_v P(v|u) \log(P(\tau^{\text{real}} = v|u)) \\
&= \mathcal{H}(\tau^{\text{real}}) + \sum_{u,v} P(u,v) \log(P(v|u)).
\end{aligned}
\tag{8}
$$

To reduce computational complexity, we alternately optimize the total mutual information with respect to world model and policy network. First, we fix the policy parameters $\theta$ and only optimize the parameters of world model $\psi$ to maximize the total mutual information $I_{\theta,\psi}(\tau^{\text{img}}, \tau^{\text{real}})$. Since the first term $\mathcal{H}(\tau^{\text{real}})$ measures the entropy of real trajectories generated by policy $\pi_\theta$ on real MDP, it is not related to parameters of the world models $\psi$ and we can remove this term. As for the second term $\sum_{u,v} P(u,v) \log(P(v|u))$, we consider the fact that our world model in conjunction with the policy network, can be regarded as a predictor for real trajectories and the second term serves as a log likelihood of a real trajectory of given imagined one. Thus, optimizing this term is equivalent to minimize the prediction error on training pairs of imagined and real trajectories $(u, v)$. When the policy is fixed, $P(u, v)$ is tractable and we can directly approximate it by sampling the data from replay buffer $\mathcal{B}$ (i.e., a collection of experienced trajectories). Thus, the second term becomes $\sum_{u,v \sim \mathcal{B}} \log(P(v|u; \psi))$, which is equivalent to the conventional model prediction error $-\mathcal{L}_\psi^{\text{Model}}$. In summary, we can get the gradient,

$$
\nabla_\psi I_{\theta,\psi}(\tau^{\text{img}}, \tau^{\text{real}}) = -\nabla_\psi \mathcal{L}_\psi^{\text{Model}}(\tau^{\text{img}}, \tau^{\text{real}}),
\tag{9}
$$

Second, we fix the model parameters $\psi$ and only optimize the parameters of policy network $\theta$ to maximize the total mutual information $I_{\theta,\psi}(\tau^{\text{img}}, \tau^{\text{real}})$. The first term of mutual information becomes maximizing the entropy of the current policy. In some sense, this term encourages exploration and also learns a robust policy. We use a Gaussian distribution $\mathcal{N}(m_\theta(s_t), v_\theta(s_t))$ to model the stochastic policy $\pi_\theta$, and thus can analytically compute its entropy on real data as $\mathbb{E}_{s_t \sim \tau^{\text{real}}} \frac{1}{2} \log 2\pi e v_\theta^2(s_t)$. Then we consider how to optimize the second term, $\sum_{u,v} P(u,v) \log(P(v|u))$. The joint distribution of real and imagined trajectories $P(u, v)$ is determined by the policy $\pi_\theta$. When the updates of the world model are stopped, the log likelihood of a real trajectory of given imagined one $\log(P(v|u))$ is fixed and can be regarded as the weight for optimizing distribution $P(u, v)$ by policy. Thus, the essential objective of maximizing $\sum_{u,v} P(u,v) \log(P(v|u))$ with respect to policy parameters $\theta$ is to guide policy to the space with high confidence of model prediction (i.e., high log likelihood $\log(P(v|u))$). Specifically, we implement it by a confidence-aware policy optimization, which reweights the degree of learning by prediction confidence $\log(P(\tau^{\text{img\_roll}}|\tau^{\text{img}}))$ during the policy improvement process. The new objective of reweighted policy improvement is written as $\log(P(\tau^{\text{img\_roll}}|\tau^{\text{img}}))\mathcal{J}_\theta^{\text{SVG}}(\tau^{\text{img\_roll}})$. In addition, we normalize the confidence weight for each batch to make training stable. In summary, the gradient of policy optimization is rewritten as:

$$
\begin{aligned}
&\nabla_\theta \left( I_{\theta,\psi}(\tau^{\text{img}}, \tau^{\text{real}}) + \mathcal{J}_\theta^{\text{SVG}}(\tau^{\text{img}})) \right) \\
=&\nabla_\theta \left( \mathbb{E}_{s_t \sim \tau^{\text{real}}} \frac{1}{2} \log 2\pi e v_\theta^2(s_t) + \log(P(\tau^{\text{img\_roll}}|\tau^{\text{img}}))\mathcal{J}_\theta^{\text{SVG}}(\tau^{\text{img\_roll}}) \right).
\end{aligned}
\tag{10}
$$

From Equation 9 and 10, we can see there are three terms, model error minimization, policy entropy maximization, and confidence-aware policy optimization, derivated by our total objective of optimizing mutual information between real and imaginary trajectories. We have the same model error loss as Dreamer, and thus the main difference from Dreamer is the policy entropy maximization and confidence-aware policy optimization. Intuitively, entropy maximization term aims at increasing the search space of SVG-based policy search like Dreamer and thus can explore more possibilities. Then the confidence-aware optimization term reweighs the search results by confidence, which contributes to improve the search quality and make sure the additional search results from large entropy are reliable enough. This approach has strong connections to distributional shift refinement in offline RL setting and may be beneficial to the community of batch RL [57]. In addition, considering that $\tau^{\text{real}}$, $\tau^{\text{img}}$ and $\tau^{\text{img\_roll}}$ are trajectories under current policy, we use a first-in-first-out replay buffer with limited capacity to mimic a approximately on-policy data stream.

Algorithm 1 summarizes our entire algorithm of optimizing mutual information and policy.

**Algorithm 1** BIRD Algorithm
---
Initialize buffer $\mathcal{B}$ with random agent.
Initialize parameters $\theta, \phi, \psi$ randomly. Set hyper-parameters: imagination horizon $H$, learning step $C$, interacting step $T$, batch size $B$, batch length $L$.
**while** not converged **do**
    **for** $i = 1 \ldots C$ **do**
        Draw $B$ data sequences $\{(o_t, a_t, r_t)\}_t^{t+L}$ from $\mathcal{B}$.
        Compute latent states $s_t \sim p_\psi(s_t|s_{t-1}, a_{t-1}, o_t)$ and imaginary trajectories $\{(s_x, a_x)\}_{x=t}^{t+H}$
        For each $s_x$, predict rewards $p_\psi(r_x|s_x)$ and values $v_\phi(s_x)$ ▷ *Calculate imaginary returns*
        Update $\theta, \phi, \psi$ using Equation 5         ▷ *Optimize policy and mutual information*
    **end for**
    Reset $o_1$ in real world.
    **for** $t = 1 \ldots T$ **do**
        Compute latent state $s_t \sim p_\psi(s_t|s_{t-1}, a_{t-1}, o_t)$.
        Compute $a_t \sim \pi_\theta(a_t|s_t)$ using policy network and add exploration noise.
        Take action $a_t$ and get $r_t, o_{t+1}$ from real world.         ▷ *Interact with real world*
    **end for**
    Add experience $\{(o_t, a_t, r_t)_{t=1}^T\}$ to $\mathcal{B}$.
**end while**
---

### 4.3 Policy Optimization with Entropy Maximization

In the context of model-free RL, maximum entropy deep RL [49, 58] contributes to learning robust policies with estimation errors, generating a question: if we simply add a maximization objective for policy entropy in the context of model-based RL with stochastic value gradients, can we also obtain policies from imaginations that generalize well to real environment? Thus, we design an ablation version of BIRD, Soft-BIRD, which just adds a entropy augmented objective to the return objective:

$$\pi_\theta^* = \arg\max_\theta \sum_t \mathbb{E}\left(r_t + \alpha \mathcal{H}(\pi(\cdot|s_t))\right), \tag{11}$$

where $\alpha$ is a hyper-parameter. We use a soft Bellman Equation for value function $v_\phi'(s_t)$ like SAC [49] and rewrite the objective of policy improvement $\mathcal{J'}_\theta^{\text{SVG}}$ as:

$$v_\phi'(s_t) = \mathbb{E}\left(r_t + \alpha \mathcal{H}(\pi_\theta(\cdot|s_t)) + \gamma v_\phi'(s_{t+1})\right),$$

$$\mathcal{J'}_\theta^{\text{SVG}}(\tau^{\text{img}}) = \mathbb{E}_{a_i \sim \pi_\theta, s_i \sim p_\psi(s_i|s_{i-1}, a_{i-1})} \sum_{k=1}^{H} \lambda_k \left[\left(\sum_{i=t}^{h-1} \gamma^{i-t}(r_i + \alpha \mathcal{H}(\pi_\theta(\cdot|s_i)))\right) + \gamma^{h-t} v_\phi'(s_h)\right]. \tag{12}$$

Compared to BIRD, soft-BIRD only maximizes the entropy of the policy instead of optimizing the mutual information between real and imaginary trajectories generated from the policy, which will provide further insights on the contribution of BIRD.

## 5 Experiments

We evaluate BIRD on DeepMind Control Suite (`https://github.com/deepmind/dm_control`) [30], a standard benchmark for continuous control. In Section 5.2, we compare BIRD with both model-free and model-based RL methods. For model-free baselines, we compare with D4PG [59], a distributed extension of DDPG [2], and A3C [56], the distributed actor-critic approach. We include the scores for D4PG with pixel inputs and A3C with state inputs, which are also used as baselines in Dreamer. For model-based baselines, we use PlaNet [12] and Dreamer [13], two state-of-the-art model-based RL. Some popular model-based RL papers [60, 61, 62, 63] are not inlcuded in our experiments since they use MPC for sampling-based planning and do not show effectiveness on RL tasks with image inputs. Compared to the MPC-based approaches that generate many rollouts to select the highest performing action sequence, our paper builds upon analytic value gradients that can directly propagate gradients through a differentiable world model and is more computationally efficient on domains that require learning from pixels. Our paper focuses on visual control tasks, and thus we only compare with state-of-the-art algorithms of these tasks (i.e., PlaNet and Dreamer).

In addition, we conduct an ablation experiment in Section 5.3 to illustrate the contribution of mutual information maximization. In Section 5.4, we further study cases and visualize BIRD's generalization to real-world information.

## 5.1 Experiment Setting

We mainly follow the experiment settings of Dreamer. Among all environments, observations are $64 \times 64 \times 3$ images, rewards are scaled to 0 to 1, and the dimensions of action space vary from 1 to 12 . Action repeat is fixed at 2 for all tasks. We implement Dreamer by its released codes (https://github.com/google-research/dreamer) and all hyper-parameters remain the same as reported. Since our model loss term in Equation 9 has the same form as Dreamer, we directly use the same model learning component as Dreamer that adopts multi-step prediction and removes latent overshooting used in PlaNet. We also use the same architecture for neural networks thus we have the same computational complexity as Dreamer. Specifically, CNN layers are employed to compress observations into latent state space and GRU [64] is used for learning latent dynamics. Policy network, reward network, and value network are all implemented with multi-layer perceptrons (MLP) and they respectively trained with Adam optimizer [65]. For all experiments, we select a discount factor of 0.99 and a mutual information coefficient of 1e-8. Buffersize is 100k. We train BIRD with a single Nvidia 2080Ti and a single CPU, and it takes 8 hours to run 1 million samples.

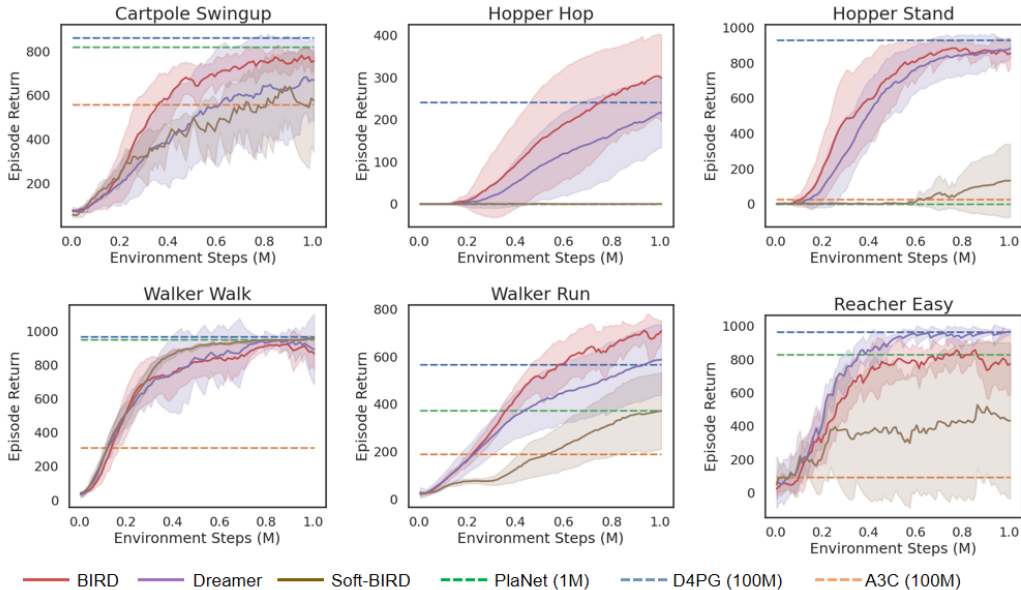

Figure 2: Results on DeepMind Control Suite. The shaded areas show the standard deviation across 10 seeds. BIRD achieves considerable performance in several challenging tasks and requires less samples than baselines.

## 5.2 Results on DeepMind Control Suite

Learning policy from raw visual observation has always been a challenging problem for RL algorithms. We significantly improve the state-of-the-art visual control approach on the visual control tasks from DeepMind Control Suite, which provides a promising avenue for model-based policy learning from pixels. Figure 5 shows the training curves on 6 tasks and additional results are placed in supplementary materials. Comparison results demonstrate that BIRD significantly outperforms baselines in terms of sample efficiency. We observe that BIRD can use half training samples to obtain the same score with PlaNet and Dreamer in *Hopper Stand* and *Hopper Hop*. Among all tasks, BIRD achieves comparable performance to D4PG and A3C, which are trained with 1,000 times more samples. In addition, BIRD achieves higher or similar convergence scores in all tasks than baselines. Here, we provide insights into the superiority of BIRD. As the mutual information between real and imaginary

trajectories increases, the behaviors that BIRD learns using the world model can be adapted to the real environment more appropriately and faster, while other model-based methods require a slower adaptation process. Besides, although world model usually tend to overfit poor policies in the early stage, BIRD will not be tempted by greedy policy optimization on the poor trajectories generated by such an imperfect model. Because the entropy maximization term in Equation 10 endows the agent a stronger exploration ability, and the confidence-aware policy optimization term encourages it re-estimate all the gathered trajectories and focus on optimizing high-confidence ones.

## 5.3   Ablation Study

In order to verify the outperformance of BIRD is not simply due to simply increasing the entropy of policy, we conduct an ablation study that compares BIRD with Soft-BIRD (4.3). Figure 5 shows the best performance of Soft-BIRD, but there is still a big gap from BIRD. As shown in *Walker Run* of Figure 5, we find that the score of Soft-BIRD first rises for a while, but eventually falls. The failure of Soft-BIRD suggests that policy improvement in model-based RL with analytic gradients is bottlenecked by the discrepancy of reality and imagination, thus only improving the entropy of policy will not help.

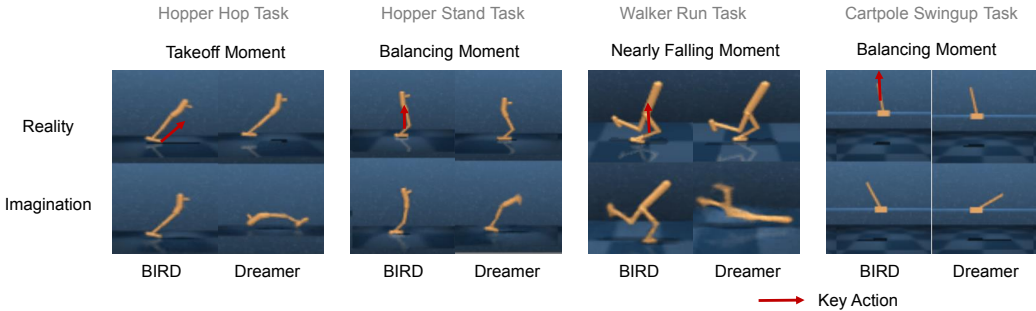

Figure 3: Prediction comparison between BIRD and Dreamer. BIRD has better predictions on key actions that will have long-term impacts, which implies that the policy generalizes to real environments well.

## 5.4   Case Study: Predictions on Key Actions

Our algorithm learns a world model with better generalization to real trajectories, especially on key actions which matters for long-horizon behavior learning. We visualize some predictions on key actions, such as the explosive force for standing up and jumping in *Hopper Stand* and *Hopper Hop*, stomping with front leg to prevent tumble in *Walker Run*, and throwing pole up to keep stable in *Cartpole Swingup*. As shown in Figure 3, BIRD makes more accurate predictions compared to Dreamer. For example, in *Hopper Hop*, Dreamer wrongly predicts the takeoff moment to fall down while BIRD has an accurate foresight that the agent will leap from the ground. Precise forecast of the key actions implicitly suggests that our imaginary trajectories generated by the learned policy indeed possess more real-world information.

## 6   Conclusion

Generalization from imagination to reality is a crucial yet challenging problem in the context of model-based RL. In this paper, we propose a novel model-based framework, called **Br**I**dging **R**eality and **D**ream (**BIRD**), which not only performs differentiable planning on imaginary trajectories, but also encourages adaptive generalization to reality by optimizing mutual information between imaginary and real trajectories. Results on challenging visual control tasks demonstrate that our algorithm achieves state-of-the-art performance in terms of sample efficiency. Our ablation study further shows that the superiority is attributed to maximizing mutual information rather than simply increasing the entropy of the policy. In the future, we will explore directions to further improve the generalization of imaginations, such as generalizable representations and reusable skill discovery.

## Broader Impact

Model-free RL requires a large amount of samples, thus limits its applications to real-world tasks. For example, the trial-and-error training process of a robot requires substantial manpower and financial resources, and certain harmful actions can greatly reduce the life of the robot. Building a world model and learning behaviors by imaginations provides a boarder prospect for real-world applications. This paper is situated in model-based RL and further improves sample efficiency over existing work, which will accelerate the development of real-world applications on automatic control, such as robotics and autonomous driving. In addition, this paper tackles a valuable problem about generalization, from imagination to reality, thus it is also of great interest to researchers in generalizable machine learning.

In the long run, this paper will improve the efficiency of factory operations, avoid artificial repetition of difficult or dangerous work, save costs, and reduce risks in the industrial and agricultural industry. For daily life, it will create a more intelligent lifestyle and improve the quality of life.

Our algorithm is a generic framework that does not leverages biases in data. We evaluated our model in a popular benchmark of visual control tasks. However, similar to a majority of deep learning approaches, our algorithm has a common disadvantage. The learned knowledge and policy is not friendly to humans and it is hard for us to know why the agent learns to act so well. Interpretability has always been a challenging open question and in the future we are interested in incorporating recent deep learning progresses on causal inference into RL.

## Acknowledgments and Disclosure of Funding

This work is supported in part by Science and Technology Innovation 2030 – "New Generation Artificial Intelligence" Major Project (No. 2018AAA0100904), and a grant from the Institute of Guo Qiang,Tsinghua University.

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
