[Reviews · NeurIPS 2020]

Review 1

Summary and Contributions: This work proposes an algorithm, BIRD, that addresses the issue in current MBRL methods where the policy or value function is updated with incorrect trajectories from a learned model. They propose to incorporate an additional objective that maximizes the mutual information between the model predicted trajectory and the real on-policy trajectory to fix this issue. This objective consists of two terms, one which maximizes the entropy of the policy, and another that minimizes model error.

Strengths: The authors show the derivation of the objective in a very clear way, of incorporating a mutual information maximization between the model output trajectories and true trajectories. There are compelling results comparing performance in 6 DMC tasks, showing that BIRD performs better than baselines like Planet and Dreamer. The flow diagrams and pseudocode make very clear the modifications to the Dreamer algorithm, and there is a nice, very thorough related work section that breaks down the different types of MBRL methods. Figure 3 is a good, qualitative example of how their method learns a more accurate model for key actions compared to Dreamer.

Weaknesses: While I find this paper reasonably thorough, I'm skeptical of the novelty. It seems the two components that differentiate it from Dreamer come from this mutual information maximization objective, which is to maximize the policy entropy and minimize the model loss. While there is an ablation showing what happens if you remove the model loss component, there is no ablation showing what happens if you remove the entropy maximization. My assumption is that the core reason for improvement is the model loss, which is not a surprising result. Doing this ablation would address this concern. It is also unclear to me how the model loss component differs from the original Dreamer objective to train the latent transition model? Is it that it is matching an entire trajectory of data instead of 1-step transitions pulled from a replay buffer? Would this make it more similar to the latent overshooting used in the Planet paper? More clarification here would be helpful.

Correctness: Yes.

Clarity: Yes, the paper is clear. There are typos: in abstract: "pone to be sucked in an inferior local policy" Unclear as to the meaning of this phrase in L251: "thus only improving the entropy of policy will not help as math."

Relation to Prior Work: Yes, the related work is thorough.

Reproducibility: Yes

Additional Feedback: I have read the rebuttal. The authors have confirmed that the only component of BIRD that differs from Dreamer is the policy entropy maximization term. While the derivation of this objective and the model loss minimization coming from optimizing mutual information between real and imaginary trajectories is interesting, I don't think it is particularly novel. The additional results on more tasks in Fig 1 of the rebuttal also do not provide significantly improved results over Dreamer. I am keeping my score as is.


Review 2

Summary and Contributions: This paper considers the issue of discrepancies between imagined and real trajectories in model-based value estimation and policy search. The authors propose to overcome this by using a mutual information maximization objective to improve the prediction of imagine trajectories such that they are close to real trajectories. The experiments in the paper show that adding this mutual information maximization objective to the state-of-the-art model-based RL algorithm Dreamer results in improved performance.

Strengths: - The proposed objective is simple and can be easily used in existing model-based RL algorithms. - The experiments demonstrate that the proposed objective leads to better performance. - The ablation studies demonstrate that the proposed objective leads to qualitatively better predictions.

Weaknesses: - Even though the experiments are performed on top of Dreamer, the paper only presents results for a subset of the tasks considered in the Dreamer paper. The results for the higher-dimensional tasks such as cheetah and quadruped are not presented. - Regularization of model-based RL so that the imagined and real trajectories are similar have also been identified and considered in several other papers. For example ensembling [1], DAE regularization [2], energy-based models [3], and Section 3.6 in [4]. The paper does not compare to any of them.

Correctness: Yes, the empirical methodology is same as in Dreamer.

Clarity: Yes, the paper is well written and easy to read.

Relation to Prior Work: The experiments in the paper build upon Planet and Dreamer methods and the relation to them is clearly discussed. The problem of accumulating model errors have been considered in several other recent works and this is not discussed.

Reproducibility: Yes

Additional Feedback: Although the qualitative ablation studies in the paper are interesting, a rigorous comparison (with Dreamer) of prediction error such as plots of multi-step prediction error would further support the paper. Minor comments: - Remove acquire in L49? - Typo in L75: model-based - Hopper Stand mentioned twice in L234 - Typo in L251: help as math References: [1] Deep Reinforcement Learning in a Handful of Trials using Probabilistic Dynamics Models. Chua et al. (NeurIPS 2018) [2] Regularizing Trajectory Optimization with Denoising Autoencoders. Boney et al. (NeurIPS 2019) [3] Regularizing Model-based Planning with Energy-Based Models. Boney et al. (CoRL 2019) [4] Model-based Reinforcement Learning: A Survey. Moerland et al. (Arxiv, 2020) Update after author response: I acknowledge that I have read the rebuttal. The authors have satisfactorily addressed my concerns and I have update my score above the acceptance threshold.


Review 3

Summary and Contributions: [Bridging Imagination and Reality for Model-Based Deep Reinforcement Learning] In this paper, the authors propose an interesting pipeline where model-free policy updates and model-based updates can be combined. More specifically SVG is used as the model-based part and D4PG is used as the model-free part. An additional loss is used to constraint the imaginary trajectories from deviating from the real trajectories.

Strengths: 1) The algorithm is novel and interesting. While the idea of combining model-free and model-based training is straight-forward, it has been unclear how to do it efficiently and effectively. I believe it can be beneficial to the community of reinforcement learning. And it is also mathematically and engineeringly neat, making it reproducible and without the worry of heavy tuning. 2) The performance is improved with the proposed algorithms. Some of the state-of-the-art baselines, including PlaNet and Dreamer are used as comparisons. 3) The paper is well-written and easy to understand.

Weaknesses: 1) It is unclear how important SVG and D4PG are to the proposed algorithm. The ablation on the choice of sub-modules in this algorithm is missing. 2) There are also no experiments which take states as input. Some experiments are quite challenging, for example humanoid. Additionally what happens if we do not include the mutual information term?

Correctness: Yes

Clarity: Yes

Relation to Prior Work: Yes

Reproducibility: Yes

Additional Feedback: I tend to vote for acceptance to this paper, given that the contribution of the algorithm which combines model-free and model-based, and improves sample efficiency.


Review 4

Summary and Contributions: The authors present a method for using rollouts from a learned model to help increase the sample efficiency of online deep reinforcement learning.In contrast to some prior approaches, the authors explicitly use a term for model fitting to prioritize maximizing the mutual information between roll outs under the learned model and trajectories obtained from roll outs in the real world. The authors compare to DREAMER on 6 of the DeepMind Control suite tasks and see encouraging performance.

Strengths: Using models can be a promising way to increase sample efficiency in deep reinforcement learning but if the models are inaccurate it has the potential to yield worse performance. The authors propose an objective that both tries to maximize the policy performance and maximize the mutual information between the trajectories generated by running the policy in real life and the trajectories that would be generated under the learned model. The empirical results on a number of common benchmarks in the DeepMind control suite show their approach has equal or better performance than other benchmarks. Their algorithm did better on hopper than dreamer in terms of sample efficiency and slightly better (though the confidence intervals looked to overlap). These are much more efficient than some other prior approaches The authors present some nice ablation studies to help understand if the mutual information significantly impacts the algorithm, which it does.

Weaknesses: The primary contribution is empirical. Therefore I’d expect a bit more significant improvement over other recent approaches: the empirical performance seems good but not substantially better than Dreamer except on Hopper.

Correctness: The approach seems reasonable.

Clarity: Yes

Relation to Prior Work: Reasonably. There's a very long history of model based approaches in both controls and reinforcement learning, and I think that discussing things in terms of imagination, while evocative, can often minimize the connection to these other long standing ideas. Of course I agree that being able to differentiate and update the model through part of a joint objective which is focused on maximizing the resulting performance is a very interesting development in DRL.

Reproducibility: Yes

Additional Feedback: I'd like to see more than 3 seeds in figure 2, at least for Dreamer and Bird. It is hard for me to know whether there are enough details to reproduce the work without actually trying to do so. -- Post reading reviews and author rebuttal. Thanks for sharing the additional experiments. Unfortunately they actually reduce my favor of the paper a little bit-- the confidence intervals overlap substantially with Dreamer almost everywhere, and the means are very close throughout. I think the text in the rebuttal is overstating to say that Bird outperforms Dreamer on these settings. That said, Bird does have performance improvements in some places, though I'd really like to see more than 3 seeds, especially since there seems like there is huge variability. My score stays the same.

[Author Response · NeurIPS 2020]

**We thank all reviewers for their efforts and thoughtful comments, which are helpful for improving our paper.**

**Response to Reviewer #1:   Q1. Ablation on entropy loss and clarification on model loss.**   Policy entropy
maximization and model error minimization are deduced by our objective of optimizing mutual information between
real and imaginary trajectories. Since the model loss term has the same form as Dreamer, we directly use the same
model learning component as Dreamer that adopts multi-step prediction and removes latent overshooting used in PlaNet.
Our main difference from Dreamer is the policy entropy maximization and thus our comparison experiments with
Dreamer can be seen as the ablation on the entropy loss, as shown in Section 5.2. We will add these clarifications to our
paper and further refine the presentation of entropy loss and model loss. In addition, we also conduct ablation in Section
4.3 and 5.3 (BIRD vs. soft-BIRD) to show that simply encouraging policy entropy by incorporating soft learning into
model-based RL does not work.

**Response to Reviewer #2: Q1. Cheetah and Quadruped are not presented.**   As shown in Figure 1, BIRD also
outperforms Dreamer on high-dimensional (Cheetah and Quadruped) or sparse-reward tasks (Cartpole Swingup Sparse).

Figure 1: Results on more tasks.          Figure 2: Comparison of model error.

**Q2. Some papers regularizing the discrepancy between imagined and real trajectories are not discussed or**
**compared.**   These related papers use MPC (described in Line 29-41 of our paper) for sampling-based planning
and do not show effectiveness on RL tasks with image inputs. Compared to the MPC-based approaches that generate
many rollouts to select the highest performing action sequence, our paper builds upon analytic value gradients that can
directly propagate gradients through a differentiable world model and is more computationally efficient on domains that
require learning from pixels. Our paper focuses on visual control tasks, and thus we only compare with state-of-the-art
algorithms of these tasks (i.e., PlaNet and Dreamer). We will properly cite all these papers as the reviewer suggested
and add these discussions in the next version.

**Q3. A rigorous comparison (with Dreamer) of prediction error.**   Image reconstruction error will be dominated
by image background and cannot reflect the prediction error on latent state. Thus we calculate the model error as the
discrepancy between latent states that predicted by model and encoded from posterior image observations. As shown in
Figure 2, BIRD that significantly outperforms Dreamer has a much lower model error.

**Response to Reviewer #3:   Q1. It is unclear how important SVG and D4PG are to the proposed algorithm**.
Because our proposed framework adopts an end-to-end model-based RL paradigm, as a differentiable model-based
policy optimization method, SVG is an indispensable module of our framework, which accounts for the reason we
cannot remove SVG to conduct ablation study on it. In addition, our framework does not include D4PG module so we
also cannot conduct ablation on D4PG. We only use it as a baseline to demonstrate that our proposed algorithm can
significantly outperform a popular model-free RL algorithm.

**Q2. There are also no experiments which take states as input.**   This paper focuses on visual control tasks and
aims at improving the state-of-the-art RL algorithm (Dreamer) for these tasks, and thus we follow experiment settings
used by Dreamer.

**Q3. What happens if we do not include the mutual information term?**   Mutual information term consists of
model error and policy entropy. If we only remove policy entropy, our algorithm becomes Dreamer. The comparison
experiments in Section 5.2 show that our model outperforms Dreamer. If we further remove model error, the performance
of learned policy will be much worse.

**Response to Reviewer #5:   Q1. I'd expect a bit more significant improvement over recent approaches.**   Learn-
ing policy from raw visual observation has always been a challenging problem for RL algorithms. We significantly
improve the state-of-the-art visual control approach (i.e., Dreamer) on Hopper Stand and Hopper Hop and achieve
slightly better or comparable performance on other 10 tasks by stochastic value gradients in conjunction with mutual
information maximization, which provide a promising avenue for model-based policy learning from pixels.

**Q2. More seeds.**   We run experiments with more seeds and the results are similar. We will add these results in the
next version.

[Meta-Review · NeurIPS 2020]

The paper introduces the BIRD algorithm, a model-based RL algorithm based on differentiable planning (SVG-like). A key aspect of BIRD is a Mutual Information term in the loss function, which encourages the similarity of the imaginary data and the real observations. Reviewers generally liked this paper, even though there have been some concerns related to the extent of its novelty, especially compared to Dreamer. I summarize some of the concerns here, which should be addressed in the revised version of this work. Please refer to the reviews for more detail, and revise your paper by incorporating their comments. - This paper has some similarities to Dreamer. If we expand the MI, the main difference with Dreamer would be the existence of policy entropy term. It is important that the authors expand on this and clearly state what differentiate this work with Dreamer. - The number of runs (3) in experiments is too few. There is a large overlap between confidence intervals, and in some cases it is difficult to say whether this algorithm is better than alternatives. Please increase the number of runs to a much larger value, such as 10 or 20. - It is encouraged to include the results of other high-dimensional tasks.